# Collective dynamics and long-range order in thermal neuristor networks

Yuan-Hang Zhang [1] ✉, Chesson Sipling [1], Erbin Qiu[1,2], Ivan K. Schuller [1] & Massimiliano Di Ventra [1] ✉

In the pursuit of scalable and energy-efficient neuromorphic devices, recent research has unveiled a novel category of spiking oscillators, termed "thermal neuristors." These devices function via thermal interactions among neighboring vanadium dioxide resistive memories, emulating biological neuronal behavior. Here, we show that the collective dynamical behavior of networks of these neurons showcases a rich phase structure, tunable by adjusting the thermal coupling and input voltage. Notably, we identify phases exhibiting long-range order that, however, does not arise from criticality, but rather from the time non-local response of the system. In addition, we show that these thermal neuristor arrays achieve high accuracy in image recognition and time series prediction through reservoir computing, without leveraging long-range order. Our findings highlight a crucial aspect of neuromorphic computing with possible implications on the functioning of the brain: criticality may not be necessary for the efficient performance of neuromorphic systems in certain computational tasks.

Neuromorphic computing, a field inspired by brain functionality, represents a powerful approach to tackle a wide range of information processing tasks that are not instruction-based, such as those typical of artificial intelligence and machine learning[1–3]. Unlike traditional computers that use the von Neumann architecture, separating memory and computing, neuromorphic systems utilize artificial neurons and synapses. These components can be implemented using diverse physical systems, such as photonics[4], spintronics[5], resistive switching materials[6,7], and electrochemical devices[8].

In neuromorphic systems, regardless of the underlying physical framework, information processing is executed via a spiking neural network[9]. Neurons in this network emit spikes in response to specific external stimuli. These spikes travel through synapses, either exciting or inhibiting downstream neurons. During training for a particular task, synaptic weights are iteratively updated, guided by either biologically-inspired algorithms like spike timing-dependent plasticity[10] and evolutionary algorithms[11] or adaptations of traditional machine learning algorithms like backpropagation[12].

The collective, as opposed to the individual behavior of the neurons in the network, facilitates the aforementioned tasks. This collective behavior may also be essential for the functioning of the animal brain. For instance, the critical brain hypothesis suggests that the brain operates in a state of criticality; namely, it is poised at a transition point between different phases[13–17]. This critical state is believed to be optimal for the brain's response to both internal and external stimuli, due to its structural and functional design. Yet, despite the popularity of the hypothesis, questions and doubts remain, and some argue that the brain is not truly critical or not critical at all[15,18–20].

In our present study, we do not aim to directly tackle the critical brain hypothesis. Rather, we approach the subject from a different angle: we examine a neuromorphic system that exhibits brain-like features. With similar working principles, one can then naturally extend the critical brain hypothesis to neuromorphic systems and question whether spiking neural networks also function at a critical state. This topic remains contentious, and arguments supporting[21,22] and opposing[23] the notion have been reported, each presenting slightly different definitions and perspectives.

In this work, we show that a neuromorphic system may support long-range ordered (LRO) phases, without criticality. The origin of this LRO is the time non-local (memory) response of the system to external perturbations. On the other hand, we show that such LRO is not necessary for certain computational tasks, such as classification and

[1]Department of Physics, University of California San Diego, La Jolla, CA 92093, USA. [2]Department of Electrical and Computer Engineering, University of California San Diego, La Jolla, CA 92093, USA. ✉e-mail: yuz092@ucsd.edu; diventra@physics.ucsd.edu

time series predictions. These results may provide some hints on the functioning of biological brains.

As a specific example, we consider a neuromorphic system comprised of thermal neuristors[7,24], based on vanadium dioxide ($VO_2$) spiking oscillators that communicate via heat signals. The properties of the individual oscillators (which take advantage of the hysteric metal-insulator transition of $VO_2$) and their mutual interactions have been experimentally validated earlier[7,24]. These earlier studies form the basis of our numerical model of a large-scale network, which allows us to numerically analyze the collective dynamics of the system. We find that the different phases can be tuned by varying the thermal coupling between the neurons and the input voltage. We apply this system to image recognition tasks using reservoir computing[25] and explore the relationship between performance and collective dynamics. We find that LRO does not necessarily enhance the performance in tasks like image recognition, a result in line with the findings of ref. 23.

## Results

$VO_2$-based oscillators have been utilized as artificial neurons in many previous studies[7,24,26–30], each featuring slightly different designs, mechanisms, and applications. In particular, we focus on thermal neuristors, a concept pioneered in ref. 7, which effectively reproduces the behavior of biological neurons. These neuristors are not only straightforward to manufacture experimentally but also exhibit advantageous properties such as rapid response times and low energy consumption.

Figure 1a presents the design and circuitry of the thermal neuristor, featuring a thin $VO_2$ film connected in series to a variable load resistor. $VO_2$ undergoes an insulator-to-metal transition (IMT) at approximately 340 K[31], with different resistance-temperature heating and cooling paths, which leads to a hysteresis loop, as depicted in Fig. 1b. Additionally, the system includes a parasitic capacitance resulting from the cable connections, which is vital for the neuristor's operation.

The behavior of the circuit displayed in Fig. 1a closely resembles a leaky integrate-and-fire neuron[32]. The capacitor $C$ is charged up by the voltage source, $V^{in}$, and slowly leaks current through $R$. When the voltage across $VO_2$ reaches a threshold, joule heating initiates the IMT, drastically reducing resistance in the $VO_2$ which causes $C$ to discharge, leading to a current spike. At the same time, the reduced resistance leads to reduced joule heating, which is then insufficient to maintain the metallic state, causing the $VO_2$ film to revert to its insulating phase. This process repeats, producing consistent spiking oscillations.

We have experimentally fabricated and evaluated this system of $VO_2$-based thermal neuristors. The spiking behavior of a single neuristor is shown in Fig. 1c. With insufficient heating, the neuristor does

not switch from the insulating state whereas excessive heating keeps it perpetually in the metallic state. As a consequence, no spiking patterns emerge when the input voltage is too low or too high. Numerical simulations, using the model described in the next section, corroborate this behavior, mirroring the experimental findings.

Distinct from biological neurons that communicate via electrical or chemical signals, thermal neuristors interact through heat. As illustrated in Fig. 1a, adjacent neuristors, while electrically isolated, can transfer heat via the substrate. Each current spike produces a heat spike, which spreads to nearby neuristors, reducing their IMT threshold voltage, thereby causing an excitatory interaction. Conversely, excessive heat can cause neighboring neuristors to remain metallic and cease spiking, akin to inhibitory interactions between neurons. Further experimental insights on neuristor interactions are detailed in Appendix B the supplementary information (SI).

Although we have experimentally shown that a small group of thermal neuristors can mirror the properties of biological neurons, effective computations require a vast network of interacting neurons. Before building a complex system with many neuristors, we first simulate a large array of thermal neuristors, providing a blueprint for future designs.

### Theoretical model

The theoretical model builds upon the framework established in ref. 7, with some minor adjustments. The system is built of identical neuristors, uniformly spaced in a regular 2-dimensional array. Their behavior is governed by the following equations:

$$C\frac{dV_i}{dt} = \frac{V_i^{in}}{R_i^{load}} - V_i\left(\frac{1}{R_i} + \frac{1}{R_i^{load}}\right), \tag{1}$$

$$C_{th}\frac{dT_i}{dt} = \frac{V_i^2}{R_i} - S_e(T_i - T_0) + S_c\nabla^2 T_i + \sigma\eta_i(t). \tag{2}$$

Equation (1) describes the current dynamics, with each variable corresponding to those shown in Fig. 1a. Equation (2) describes the thermal dynamics, including the coupling between nearest-neighbor neuristors. Here, $T_0$ represents the ambient temperature, $C_{th}$ is the thermal capacitance of each neuristor, $S_e$ denotes the thermal conductance between each neuristor and the environment, and $S_c$ refers to the thermal conductance between adjacent neuristors. $\eta_i(t)$ represents a Gaussian white noise variable for each neuristor that satisfies $\langle\eta_i(t)\eta_j(t')\rangle = \delta_{i,j}\delta(t - t')$, and $\sigma$ is the noise strength. Detailed values of

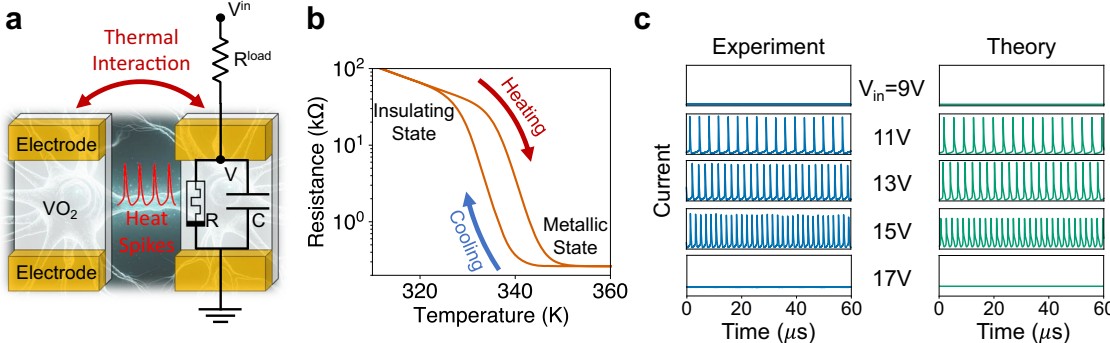

**Fig. 1 | Overview of the thermal neuristor model. a** Schematic and circuit diagram of two neighboring thermal neuristors. Each neuristor is modeled as an RC circuit, which undergoes stable spiking oscillations with proper external input. Neighboring neuristors are electrically isolated but communicate with each other through thermal interactions. **b** The resistance-temperature characteristic of the $VO_2$ film, denoted by the variable resistor R in (**a**). $VO_2$ exhibits an insulator-to-metal transition at approximately 340 K, characterized by distinct heating and cooling

trajectories, thus forming a hysteresis loop. **c** Illustration of stable spiking oscillations in a single neuristor across various input voltages, with the y-axis range for each plot set between 0 and 5 mA. Numerical simulations based on Eqs. (1) and (2) align well with experimental data, demonstrating stable spiking patterns within a certain input voltage range and an increase in spiking frequency proportional to the input voltage. Source data are provided in the Source Data file.

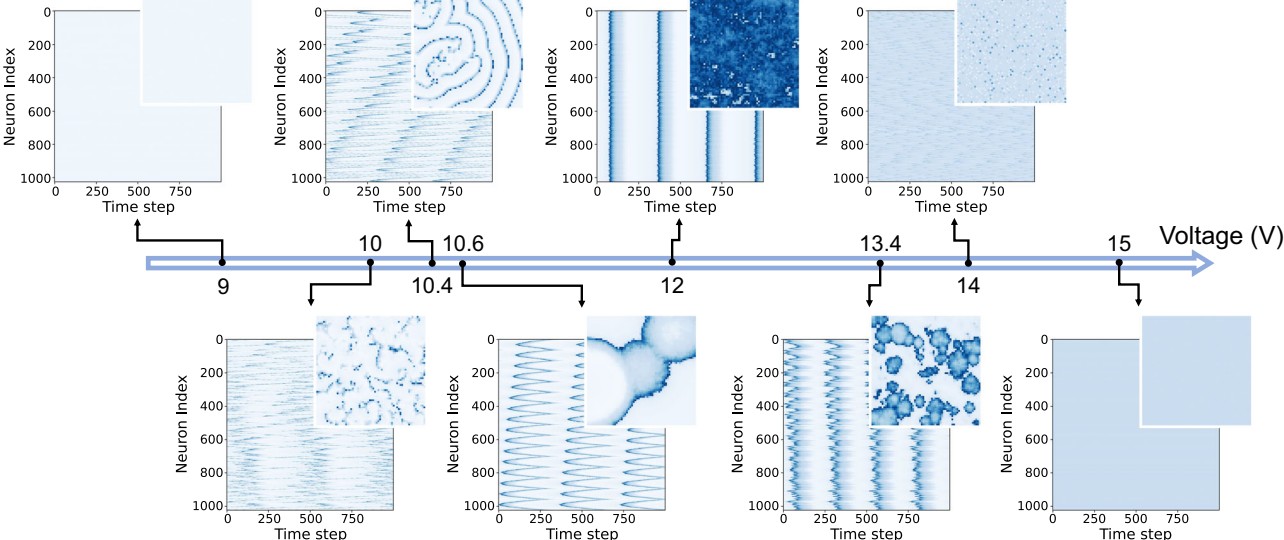

**Fig. 2 | Snapshots of different oscillation patterns in a 64 × 64 array of thermal neuristors.** In each panel, color indicates current level: white signifies no current, while shades of blue denote current spikes. The main panels show collective current-time plots for the first 1024 neuristors (concatenated from the first 16 rows), and each inset captures a specific moment in the 64 × 64 array. The system exhibits no activity at very low input voltages. As the voltage increases, a sequence of dynamic phases unfolds, including correlated clusters (10 V and 13.4 V), system-wide waves (10.4 V and 10.6 V), synchronized rigid states (12 V), and uncorrelated spikes (14 V), culminating again in inactivity at excessively high voltages. The thermal capacitance, $C_{th}$, is fixed at the experimentally estimated value. Detailed simulation parameters can be found in the methods section, and dynamic visualizations of these spiking patterns are available in Supplementary Movie 1.

these constants are provided in the methods section. $R_i$ is the resistance of the VO$_2$ film, which depends on temperature and its internal state, or memory, following the hysteresis loop depicted in Fig. 1b. This memory factor is pivotal in determining the collective behavior of thermal neuristors. We utilize the hysteresis model formulated in ref. 33, with comprehensive details available in the methods section.

## Numerical results

We used the theoretical model to simulate an $L \times L$ square lattice comprised of identical thermal neuristors, whose dynamics are governed by Eqs. (1) and (2). Different input voltages $V^{in}$ produce a diverse array of oscillation patterns, as illustrated in Fig. 2. At very low (9V) or high (15V) input voltages, the system remains inactive, as found in individual neuristors. With a 12 V input voltage, synchronization develops, with nearly all neuristors spiking in unison, creating a phase of rigid states. A phase transition occurs slightly below $V^{in} = 10$ V, where clusters of correlated spikes start to form, then gradually turn into system-wide activity waves (10.4 V and 10.6 V). Another phase transition occurs slightly above $V^{in} = 13.4$ V, where the synchronized rigid oscillations start to fracture into smaller clusters until the individual spikes become uncorrelated (14 V).

## Analytical understanding

The emergence of a broad range of phases and long-range correlations in our system, despite only diffusive coupling existing between neurons, is a point of significant interest. Diffusive coupling is typically associated with short-range interactions, making the discovery of long-range correlations particularly intriguing.

It is well-established that long-range correlations can emerge from local interactions in various systems such as sandpiles[34], earthquake dynamics[35], forest fires[36], and neural activities[37]. These systems exhibit avalanches-cascades triggered when one unit's threshold breach causes successive activations-manifesting as power-law distributions of event sizes, indicative of scale-free or near scale-free behaviors.

Such spontaneously emerging long-range correlations are often described under the framework of self-organized criticality[34,35]. However, this term may be misleading. Criticality suggests a distinct

boundary, characterized by a phase above and below it, as seen in the sandpile model where an appropriately defined order parameter undergoes a second-order phase transition[38,39]. In contrast, systems like earthquakes, while displaying power-law behaviors, do not exhibit true scale-invariance[40] and can be described as undergoing continuous phase transitions without clear critical boundaries[39].

We argue that the observed LRO in our system, similar to those in systems without genuine scale-free behaviors, is induced by memory (time non-local) effects stemming from a separation of time scales: a slow external drive contrasts sharply with fast avalanche dynamics. In our system, we identified three distinct time scales: the metallic RC time ($\tau_{met} = R_{met}C \sim 187$ ns), the insulating RC time ($\tau_{ins} = R_{ins}C \sim 7.57$ $\mu$s), and the thermal RC time ($\tau_{th} = R_{th}C_{th} = C_{th}/(S_c + S_e) \sim 241$ ns). We observe that $\tau_{met} \lesssim \tau_{th} \ll \tau_{ins}$. As the spiking and avalanche dynamics are primarily controlled by $\tau_{met}$ and $\tau_{th}$, and the driving dynamics by $\tau_{ins}$, our system does exhibit an approximate separation of time scales.

This separation allows us to conceptualize the slower time scale as memory, which retains long-term information about past states and remains relatively constant within the faster time scale, capable of preserving non-local temporal correlations. As a consequence, neuristors that are spatially distant are progressively coupled, resulting in long-range spatial correlations. This concept is systematically explored in a spin glass-inspired model[41], and similar behavior is also observed in a class of dynamical systems with memory (memcomputing machines) used to solve combinatorial optimization problems[42]. In Appendix A in the SI, we provide an analytical derivation of this phenomenon using a slightly simplified version of our model.

Consequently, altering the memory strength, specifically through adjustments of the thermal time scale $\tau_{th}$ by varying $C_{th}$ (the thermal capacitance of each neuristor), should result in changes to the oscillation patterns and the presence or absence of long-range correlations. Indeed, we find that by modifying $C_{th}$, we can control the rate of heat dissipation, effectively influencing the memory's response time. Additionally, in Appendix C5 of the SI, we present another example where increasing the ambient temperature reduces the insulating RC time, thereby diminishing memory and minimizing long-range correlations.

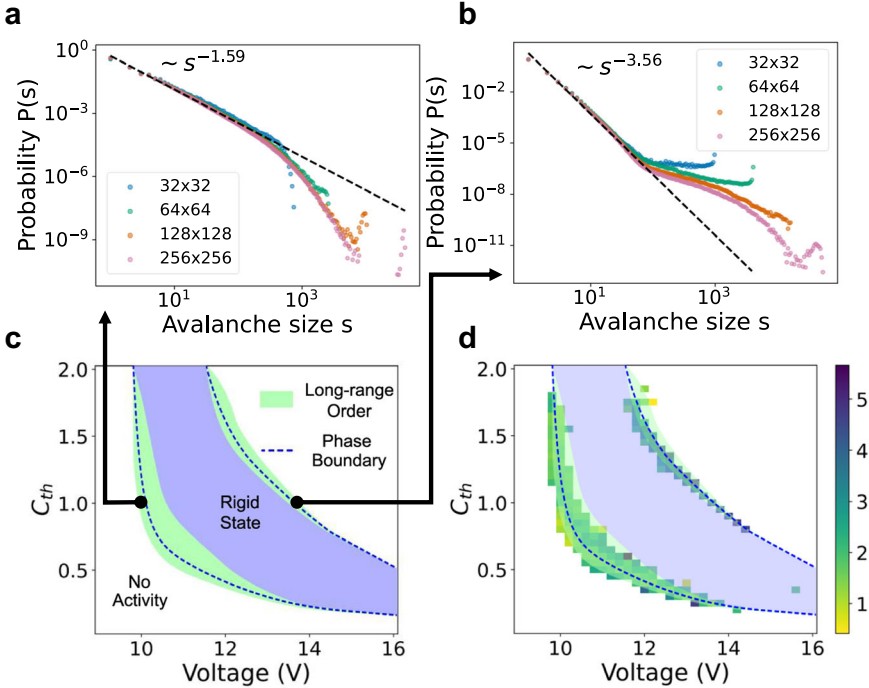

**Fig. 3 | Avalanche size distributions and phase structures in 2D thermal neuristor arrays of different sizes. a, b** Two different avalanche size distributions at phase boundaries, with both distributions obtained at $C_{th} = 1$, but different input voltages ($V^{in} = 9.96$ V for (**a**), and 13.46 V for (**b**)). **c** Phase diagram of the thermal neuristor array, with the y-axis depicting the relative value of $C_{th}$ compared to its experimentally estimated level. We observe synchronized rigid states with collective spiking and quiescent states with no spikes (no activity). Near the phase

boundaries, a robust power-law distribution in avalanche sizes is noted across various parameters, signaling the existence of LRO. **d** Exponents of the power-law fit of the avalanche size distributions (omitting the negative signs for clarity). The phase diagram from (**c**) is superimposed for enhanced visualization. Regions lacking a colored box signify a failed power-law fit, and exponents are capped at 6 to exclude outliers. Source data are provided in the Source Data file.

## Avalanche size distribution

To verify the presence of LRO in our system, we analyzed the avalanche size distribution of current spikes. Here, we define an avalanche as a contiguous series of spiking events occurring in close spatial (nearest neighbor) and temporal (400 ns) proximity. The heat generated by each spiking event transfers to the neighboring neuristors, making their IMT more likely and thus triggering a cascade of spikes. Figure 3a and b shows examples of avalanche size distributions in which a power-law distribution is observed, indicative of LRO. The methodology for identifying these avalanches is detailed in the methods section.

We varied the input voltage and thermal capacitance, $C_{th}$, to generate the phase diagram depicted in Fig. 3c. Here, the y-axis reflects $C_{th}$'s relative value against the experimentally estimated one. Similar to observations in Fig. 2, both a synchronized rigid state, characterized by collective neuristor firing, and a quiescent state, with no spiking activity, are found. Around the phase boundaries, a wide range of parameters leads to a power-law distribution in avalanche sizes across several orders of magnitude, confirming the existence of LRO. This is further supported in Fig. 3d, where we compute avalanche sizes for each point in the parameter space and plot the absolute value of the exponent from the fitted power-law distribution. Areas without a colored box indicate an unsuccessful power-law fit, with the maximum exponent limited to 6 to remove outliers.

While we empirically observe power-law scaling in avalanche sizes, one might question if this implies criticality and scale-invariance. The numerical evidence presented here suggests otherwise. First, the power-law distributions in Fig. 3a and b do not align with the finite-size scaling ansatz[39,43], which predicts diminishing finite-size effects with increasing system size. Furthermore, a rescaling based on the system size should collapse all curves onto one[21,39] for scale-invariant systems, but such an effect is notably missing in our system,

contradicting finite-size scaling expectations. In Supplementary Fig. 9, we present the results of attempted finite-size scaling, which clearly imply a lack of scale-invariance.

Despite the absence of criticality, can the system still perform some computing tasks effectively? Is the LRO observed in these thermal neuristor arrays even necessary for such tasks? We demonstrate in the following section that for classification, LRO, let alone criticality, is not necessary, as anticipated in[23].

## Role of LRO in reservoir computing classification tasks

We apply our thermal neuristor array to reservoir computing (RC) to answer the above questions. RC differentiates itself from traditional neural network models by not requiring the reservoir - the network's core - to be trained. The reservoir is a high-dimensional, nonlinear dynamical system. It takes an input signal, $\mathbf{x}$, and transforms it into an output signal, $\mathbf{y} = f(\mathbf{x})$. A simple output function, usually a fully connected layer, is then trained to map this output signal, $\mathbf{y}$, to the desired output, $\hat{\mathbf{z}} = g(\mathbf{y})$. Training typically involves minimizing a predefined loss function between the predicted output $\hat{\mathbf{z}}$ and the actual label $\mathbf{z}$, associated with the input $\mathbf{x}$, using backpropagation and gradient descent. If the output function is linear, training can be reduced to a single linear regression.

The reservoir's transfer function $f$ can be arbitrary, with its main role being to project the input signal $\mathbf{x}$ into a high-dimensional feature space. Since the reservoir doesn't require training, employing an experimentally designed nonlinear dynamical system like our thermal neuristor array for RC is both effective and straightforward.

As a practical demonstration, we applied RC using thermal neuristors to classify handwritten digits from the MNIST dataset[44]. Each 28 × 28 grayscale pixel image, representing digits 0 to 9, is converted into input voltages through a linear transformation. The system is then allowed to evolve for a specific time, during which we capture the

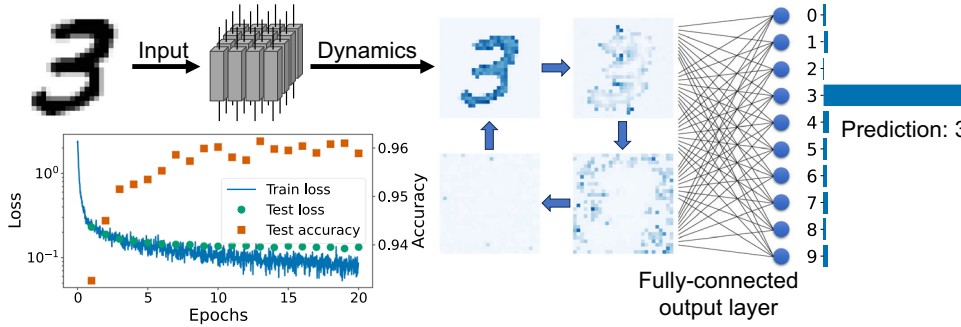

**Fig. 4 | Overview of our reservoir computing implementation with a 2D thermal neuristor array.** The MNIST handwritten digit dataset[44] is used as a benchmark. Each image from the dataset is translated into input voltages for a 28 × 28 thermal neuristor array. The array's spiking dynamics are gathered as the reservoir output. A fully connected output layer, enhanced with softmax nonlinearity, is trained to classify the digit. The bottom-left panel illustrates the training process, displaying both loss and accuracy, culminating in a final test set accuracy of 96%. Source data are provided in the Source Data file.

spiking dynamics as output features from the reservoir. Subsequently, a fully connected layer with softmax activation is trained to predict the digit. This process is schematically represented in Fig. 4.

The output layer was trained over 20 epochs, as shown in the bottom-left panel of Fig. 4. The test loss stabilized after approximately 10 epochs, and due to the network's simple architecture, overfitting was avoided. Ultimately, the test set accuracy reached 96%. Further training details can be found in the methods section.

In this experiment, we treated the voltage transformation, thermal capacitance $C_{th}$, and noise strength $\sigma$ as adjustable hyperparameters. This allowed us to check which region of phase space would produce optimal results. We found that the parameters that yielded optimal performance were an input voltage range between 10.5 V and 12.2 V, $C_{th} = 0.15$, and noise strength $\sigma = 0.2\,\mu J \cdot s^{-1/2}$. These settings placed us within the synchronized rigid phase, not the LRO one, with the input voltage variations introducing complex oscillatory patterns. In fact, choosing the parameters in the LRO phase produced worse results. We show this in Appendix C4 of the SI. The phase diagram relating to noise strength can be found in Supplementary Fig. 10, and videos of collective oscillations under image inputs are available in Supplementary Movie 2.

To further explore the role of LRO in reservoir computing tasks, Appendix C5 of the SI details our efforts to eliminate LRO within the reservoir by either removing interactions between neurons altogether or by reducing memory. We quantified LRO using the avalanche size distribution under these various settings. The findings reveal that even when the reservoir operates in a rigid or non-interacting state, long-range structures inherited from the dataset are still apparent. However, no relation between LRO and computational performance was observed. Videos demonstrating collective oscillations under these conditions are available in Supplementary Movies 3 and 4.

As further verification, Appendix C6 of the SI documents an additional experiment involving the prediction of chaotic dynamics governed by the 2D Kuramoto-Sivashinsky equations[45]. The results corroborate our primary findings: optimal performance in reservoir computing is achieved without the presence of LRO within the reservoir.

In conclusion, the spiking dynamics of the optimally performing reservoir in our study are not characterized by an LRO state. This observation aligns with the findings in[23], and challenges the well-accepted critical brain hypothesis[14] and theories suggesting that near-critical states enhance computational performance[46,47]. However, our results do not directly contradict the critical brain hypothesis, since it is possible that long-range correlations are effectively encapsulated within the feed-forward layer. Despite this possibility, our findings highlight a crucial aspect: criticality is not a prerequisite for effective computational performance in such tasks.

## Discussion

In this study, we have developed and experimentally validated $VO_2$-based thermal neuristors that exhibit brain-like features. We then formulated a theoretical model grounded in our experimental findings to facilitate large-scale numerical simulations. These simulations revealed a variety of phase structures, notably those with LRO, across a broad spectrum of parameters. Our analysis suggests that this LRO stems from the time-nonlocal response of the system and is not associated with criticality. Significantly, we demonstrate that this feature does not impair the system's computational abilities. In fact, it does not even seem to be necessary in some tasks, such as classification and time series prediction, as we have shown by using our thermal neuristor array in reservoir computing.

The thermal neuristor represents an innovative artificial neuron model, and our research offers insights into the collective dynamics of artificial neuronal activities. Our findings suggest that criticality is not a prerequisite for effective information processing in such systems. This challenges the critical brain hypothesis and its applicability to neuromorphic systems, indicating that even non-critical systems can excel in some computational tasks. We then advocate for a broader exploration of non-critical dynamical regimes that might offer computational capabilities just as powerful, if not more so, than those found at or near a critical state.

Moreover, our work highlights the potential of $VO_2$-based thermal neuristors in computing applications, setting the stage for more extensive experiments. Given the growing need for innovative hardware in neuromorphic computing, our $VO_2$-based thermal neuristor system is a promising candidate for advancing next-generation hardware in artificial intelligence.

## Methods

### Fabrication of VO₂ thermal neuristor arrays

**Epitaxial VO₂ thin film growth.** We employed reactive RF magnetron sputtering to deposit a 100-nm thick $VO_2$ film onto a (012)-oriented $Al_2O_3$ substrate. Initially, the substrate was placed in a high vacuum chamber, achieving a base pressure of around $10^{-7}$ Torr, and heated to 680 °C. The chamber was then infused with pure argon at 2.2 s.c.c.m and a gas mix (20% oxygen, 80% argon) at 2.1 s.c.c.m. The sputtering plasma was initiated at a pressure of 4.2 mTorr by applying a forward power of 100 W to the target, corresponding to approximately 240 V. Post-growth, the sample holder was cooled to room temperature at a rate of 12 °C/min. Specular x-ray diffraction analysis of the film revealed textured growth along the (110) crystallographic direction.

**VO₂ thermal neuristor arrays fabrication.** For patterning the $VO_2$ neuristor arrays, Electron Beam Lithography (EBL) was employed. Each

**Table 1 | Parameters utilized in the numerical simulations for Eqs. (1)–(3)**

| Param | Value | Physical meaning |
|---|---|---|
| $C$ | 145 pF | Capacitance |
| $R^{load}$ | 12.0 kΩ | Load resistance |
| $C_{th}$ | 49.6 pJ/K | Thermal capacitance[a] |
| $S_e$ | 0.201 mW/K | Thermal conductance to environment |
| $S_c$ | 4.11 μW/K | Thermal conductance to neighbor |
| $T_0$ | 325 K | Ambient temperature |
| $\sigma$ | 1 μJ · s$^{-1/2}$ | Noise strength |
| $R_0$ | 5.36 mΩ | Insulating resistance prefactor |
| $E_a$ | 5220 K | VO$_2$ activation energy |
| $R_m$ | 1286 Ω | Metallic resistance |
| $w$ | 7.19 K | Width of the hysteresis loop |
| $T_c$ | 332.8 K | Center of the hysteresis loop |
| $\beta$ | 0.253 | Fitting parameter in hysteresis |
| $\gamma$ | 0.956 | Fitting parameter in hysteresis |

$R^{load}$ and $T_0$ are taken from experiment, while other parameters are optimized to align the numerical model as closely as possible to the experimentally measured data.
[a]Figures show relative values to this.

neuristor, sized at 100 × 500 nm², was delineated with 500 nm gaps. The initial lithography pattern defined electrodes by depositing a 15 nm Ti layer followed by a 40 nm Au layer. To investigate thermal interactions between neuristors, a second lithography and etching step was necessary. We utilized a reactive-ion etching system to etch the exposed VO₂ films between devices, as per the second-step lithography patterns, while the negative resist shielded the electrodes and devices from etching.

**Transport measurements.** Transport measurements were conducted in a TTPX Lakeshore probe station equipped with a Keithley 6221 current source, a Keithley 2812 nanovoltmeter, a Tektronix Dual Channel Arbitrary Function Generator 3252C, and a Tektronix Oscilloscope MSO54. The current source and nanovoltmeter were utilized to gauge the device's resistance versus temperature. The Arbitrary Function Generator (AFG) was employed to apply either DC or pulse voltage bursts, while the oscilloscope monitored the output signals. Notably, the impedance for the channel assigned to measure voltage dynamics was set at 1 MΩ, and the channel for capturing spiking current dynamics was configured to 50 Ω.

## Details of numerical simulations

**Model details and constant parameters.** The constants in Eqs. (1) and (2) are crucial in our simulations, as they depend on specific experimental setups. Following the approach in[7], we optimized these parameters to closely replicate the experimental results. The chosen values are summarized in Table 1.

The resistance of the VO₂ film, $R$, is modeled based on the hysteresis model introduced in ref. 33, described by the equations:

$$R(T) = R_0 \exp\left(\frac{E_a}{T}\right) F(T) + R_m,$$

$$F(T) = \frac{1}{2} + \frac{1}{2} \tanh\left(\beta\left\{\delta\frac{w}{2} + T_c - \left[T + T_{pr}P\left(\frac{T - T_r}{T_{pr}}\right)\right]\right\}\right),$$

$$T_{pr} = \delta\frac{w}{2} + T_c - \frac{1}{\beta}[2F(T_r) - 1] - T_r,$$

$$P(x) = \frac{1}{2}(1 - \sin\gamma x)[1 + \tanh(\pi^2 - 2\pi x)].$$

(3)

Each component of Eq. (3) is detailed in ref. 33. The term $T_r$ denotes the reversal temperature, marking the most recent transition between heating and cooling processes. Here, $\delta$ equals 1 on the heating branch and -1 on the cooling branch, with all other symbols representing constant parameters. These constants were selected in accordance with[7] to accurately reflect the experimentally observed hysteresis loop, and their values are compiled in Table 1.

The noise strength $\sigma$ was chosen to facilitate a diverse range of phase structures. We conducted preliminary tests on the phase diagram by varying $\sigma$, with the results detailed in Appendix C3 in the SI.

**Numerical methods.** For the numerical integration of Eqs. (1) and (2), we employed the Euler-Maruyama method[48] with a fixed time step of $dt$ = 10 ns. The current-time trajectories were recorded, and current spikes were identified by locating the local maxima within these trajectories.

To analyze the avalanche size distribution, we first defined an avalanche as a contiguous series of spiking events occurring within a certain spatial and temporal proximity. We determined a specific window length for both spatial and temporal dimensions and then coarse-grained the spiking trajectories, categorizing each spiking event into a corresponding window. This process resulted in a $D + 1$-dimensional lattice ($D$ spatial and 1 temporal dimensions), where each lattice site denoted the number of spikes within its window. Following this, the Hoshen-Kopelman algorithm[49] was applied to identify clusters of spiking activities within the lattice. Each identified cluster was considered as one distinct avalanche, in line with our defined criteria.

The avalanche size distribution is influenced by the chosen window size. Generally, the temporal window length should be significantly longer than the duration of each spike but shorter than the interval between consecutive spikes. For all results presented in this paper, the temporal window length was set at 400 ns. In terms of spatial window length, we focused on immediate neighbors (length = 1) of each neuristor for cluster identification.

After identifying the avalanches, we computed the histogram of avalanche sizes using a logarithmic binning scheme[50], where bins are uniformly distributed on a logarithmic scale. The sizes of these bins were determined according to Scott's normal reference rule[51]. To characterize the avalanche size distributions presented in Fig. 3, we applied a power-law fit to each histogram, excluding the tails for more accurate modeling.

## Reservoir setup

In employing thermal neuristors for reservoir computing, we consider the entire neuristor array as the reservoir. The input voltages serve as the reservoir input, and the resultant spike trains are recorded as the output.

For the MNIST dataset[44], the reservoir's parameters are detailed in the main text. To record the spike trains, we simulate the system dynamics for 10 μs, extracting spikes using the method outlined in the previous section. These spike trains are then coarse-grained with a time window of $\Delta t$ = 500 ns. Each time window is assigned a binary value indicating the presence or absence of a spike. This process results in a 28 × 28 × 20 binary array representing the reservoir's spike train output. This array is then flattened into a one-dimensional sequence of 15680 elements. A fully connected layer with dimensions 15680 × 10 is trained to map the reservoir output to the ten digit classes. At the final stage, a softmax nonlinearity is applied to transform the output layer's results into predicted probabilities. Although activation functions are not typically standard in reservoir computing tasks, we still implemented the softmax activation in conjunction with negative log-likelihood loss, as it demonstrated enhanced performance compared to mean-square-error loss without an activation function.

For training this fully connected output layer, we utilized the Adam optimizer[52] with a learning rate of $10^{-3}$. The corresponding loss curve is depicted in the bottom-left panel of Fig. 4.

## Data availability

The MNIST handwritten digit dataset can be accessed at ref. 44. All experimental and raw data depicted in the figures are included in the Source Data file. Due to their large size, trajectories and snapshots of the dynamics are not stored as files but can be reproduced using the code provided at ref. 53. Source data are provided with this paper.

## Code availability

A demo code to reproduce the results in this work can be found at ref. 53.

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

## Acknowledgements
Y.-H.Z., and M.D. were supported by the Department of Energy under Grant No. DE-SC0020892. C.S. was supported by the Center for Memory and Recording Research at UCSD. E.Q. and I.K.S. were supported by the Air Force Office of Scientific Research under award number FA9550-22-1-0135.

## Author contributions
Y.-H.Z. and M.D. conceived the project. M.D. supervised the theoretical work. Y.-H.Z. did all the numerical calculations. C.S. performed the analysis on the emergence of LRO from memory. E.Q. and I.K.S. provided the experimental data. All authors have read and contributed to the writing of the paper.

## Competing interests
The authors declare no competing interests.

## Additional information

**Peer review information** : *Nature Communications* thanks Corentin Delacour and Aida Todri-Sanial, for their contribution to the peer review of this work. A peer review file is available.

