## [Peer Review File · Nature Communications]

REVIEWER COMMENTS

Reviewer #1 (Remarks to the Author):

In this manuscript, the authors have studied a computing platform based on a 2D grid of thermally coupled VO₂ oscillators. The authors have done a fine job analyzing the system's complex collective dynamics, highlighting phase transitions from an absence of oscillations to a state of synchronized oscillations by tuning the oscillator bias voltage and thermal capacitance. In particular, the authors reveal – via simulation and backed up by analytical analysis – a regime near the phase transition where the network has spatial long-range correlations. As shown by the authors, it is in this regime that the system has richer spatio-temporal dynamics and fading memory thanks to the existence of several time scales. Then, the authors simulate a 28x28 neuristor network as a reservoir for digit classification (MNIST) and observe that the system performs best in the region where oscillators are synchronized (rigid state), rather than near the phase transition region. From this experiment, the authors conclude that neither criticality nor long-range correlation is required for this particular task.

The well-written manuscript addresses an interesting topic of long-range correlation within complex dynamical systems which is very relevant for the neuromorphic and physics-based computing community.

However, my main concern is about the chosen experiment which I think should be further developed. In particular, I have the following comments:

1. Thanks to their internal fading memory, reservoir computers are generally used to process time series with the advantage that they do not require complicated training (compared to Recurrent NN). Why have the authors chosen to classify static data with a neuristor reservoir, rather than dynamical data? In my opinion, it is not so surprising that the reservoir performs best in the rigid state since the data is static, i.e. no memory is a priori needed to classify the data.
2. The authors could greatly improve the manuscript by performing a second experiment with dynamical data (such as spoken vowel recognition, time series prediction, etc.) and see if long-range order is helpful or not for this type of data.
3. I understand that using a Softmax output layer is convenient for having a probability distribution of classified labels. However, this is a highly non-linear stage that is cascaded after the reservoir. Hence, we cannot know for sure if the reservoir significantly contributes to the measured 96% accuracy (a 28x28 single-layer perceptron already achieves around 93%:

<https://www.geeksforgeeks.org/single-layer-perceptron-in-tensorflow/>). Instead, I recommend the authors train the fully connected output layer with a linear regression – as it is usually done in reservoir computing, and obtain the output label with the Argmax function if needed.

4. It would be interesting to compare the neuristor network with other existing work on thermally coupled VO2 oscillators such as:

- A Velichko et al 2020 IOP Conf. Ser.: Mater. Sci. Eng. 862 052062
(<https://iopscience.iop.org/article/10.1088/1757-899X/862/5/052062>)

- Velichko, A.; Belyaev, M.; Boriskov, P. A Model of an Oscillatory Neural Network with Multilevel Neurons for Pattern Recognition and Computing. Electronics 2019, 8, 75.
<https://doi.org/10.3390/electronics8010075>

5. Could the authors discuss the pros and cons of using thermal exchanges compared to electrical connections? Is the neuristor network limited to nearest-neighbor connections?

6. What would be the impact of the external temperature on a neuristor network? Is the system robust to ambient temperature variations?

Corentin Delacour

Reviewer #3 (Remarks to the Author):

Review of manuscript ‘Collective Dynamics and long-range order in thermal neuristor networks’

Author present a novel spiking oscillator termed ‘thermal neuristor’ based on VO2 devices. These devices are thermally coupled by sending heat spikes that perform image recognition tasks through reservoir computing.

The manuscript in the current provides several interesting concepts but in-depth descriptions are missing and overall it lacks clarity. Here are listed the comments.

1- VO2 devices have been fabricated and used as the spiking oscillator that are thermally coupled. It is not clear how is thermal coupling established. What is the minimum distance between devices for thermal coupling to occur? Also, how are devices thermally coupled on lateral and vertical directions to nearest devices assuming that an array of devices is fabricated. What is the architecture and coupling scheme (fully or sparsely coupled) among devices for the reservoir computing? Even if performed in simulation such details are relevant.

2- Thermally coupled VO2 devices have been investigated in the past. Thermal resistance coefficient to the substrate can negative impact heat buildup as heat transfer through the substrate (first shortest heat path) than to the nearest device (secondary heat path). Thus, even if there is some heat coupling, it will be a weaker effect that might not cause a change in IMT threshold voltage. It could happen that some devices are stuck in insulating state unless enough voltage is pumped to them. Then, this also kills the purpose of having a thermally coupled network because via the voltage bias the switching dynamics are predominantly controlled for each spiking VO2 oscillator. Thus, is there any understanding which is more predominant voltage or thermal coupling to the different phases that can be obtained. It could be that thermal coupling has a minor role and input voltage is the main key contributor to the phase dynamics.

3- It is stated that VO2 devices interact through heat. Can this be quantified and shown in a figure. For example, two devices coupled through heat and impact on spiking. The Y-axis can show the spike amplitude and x-axis can show the amount of heat. It will be also relevant to show the distance between two devices to achieve such amount of heat. Also, the impact of thermal conductance to the substrate should be included. For example, in Eqn.2 it is not included the substrate thermal conductance.

4- For the square LxL lattice comprised of identical VO2 thermal neuristors, it is stated that different patterns are observed with different input voltages. It is important to state (and not written in the paper), that while each oscillator is identical, each oscillator does not experience the same amount of heat. For example, the peripheral oscillators have will experience less heat than the internal oscillators in the lattice. It is because of these nonuniform thermal distributions that long-range corrections and different time scales are observed while bearing in mind that thermal transfer is a slower phenomenon than the impact of input voltage on oscillators.

5- Different time scales are also reported in the paper which is relevant, though any simulations to portray this is missing. In additional adjusting thermal scales and thermal capacitance, will directly change the VO2 spiking patterns and this is well reported in literature. The interest here is to see the collective behavior of heat and input voltage, which leads to different scale dynamics.

6- The analytical understanding section is weak (Section II.C). The different observed timescales are due to electrothermal effects and electrothermal simulations should reveal this (again missing any figure to portray this). It is not clear what authors define as 'memory' and 'long-range correlations'. These so-called 'long-range correlations' are the slow heat propagation impact on devices? An electrothermal simulation should also reveal this.

7- Section II.D is also not clear and the link with the earthquake model dynamics is not sufficiently elaborated to understand the analogy. What does 'avalanche' represent in the context of VO₂ array network? It is written that avalanche is a series of spiking events occurring at close spatial and temporal proximity. How are these defined such as close proximity, how close? Or temporal proximity, what's the timescale? If the aim is to verify the presence of long-range order effects due to heat, then the comparison of dynamics without heat vs with heat should reveal this (again electrothermal simulations should be useful here to provide insights). If the aim is to show the existence of long-range effects, a careful comparison should be used to see if indeed LROs are indeed due to the slow thermal phenomenon.

8- What does figure 3 and its subfigures represent? The explanations on Section II.D are not clear and why power-law is not fitting and what should one understand if power law is fitting or not. Thus, it is not clear how power law on these graphs are used to reveal anything meaningful.

9- Authors write 'First, the power-law distributions in Fig. 3(a)(b) do not align with the finite-size scaling ansatz [32,33], which predicts diminishing finite-size effects with increasing system size.' This sentence is not related with the rest of what's written earlier in the paper, it reads disconnected from the rest and also it is not clear why is this relevant and what's the link with the topic at hand on different scale dynamics of VO₂ array network.

10- In the same sections, it is written 'Contrary to this, the smaller system sizes (32x32, 64x64) in Fig 3(a) results in the largest avalanches, contradicting finite-size scaling expectations'. Again, this is not clear, what is the expected finite size scaling? What's this about and what does it represent. These are unclear because section II.C and II.D are not clearly written or explained.

11- Section III, experiments were performed varying thermal capacitance C_{th} and noise strength. How realistic would be to varying C_{th} knowing that this is a material property. For example, depending on the fabricated VO₂ devices, the thermal resistance and capacitance can be obtained. In addition, what is noise in this network, and how is it introduced? Secondly, what is meant by "identifying the region of phase spaces would produce optimal result", please define these. It is expected that the system would show different behavior and phase dynamics when voltage is varied. Thus, it is not clear what the authors want to show via the RC with VO₂ arrays with respect to long-range order correlations (as stated in earlier part of the paper). It is expected that

RC will provide rich phase dynamics, but what are authors aiming at to link between RC about presence/or not presence of LRO? This is not clear. Because, regardless of LRO, RC will perform some computation and extract information from highly nonlinear neurons, as this is also the aim of RC, thus, RC computation without LRO is not a surprise. For example, as a comparison proof, the same VO2 network with electrical coupling but without thermal coupling can be used for RC.

12- The main hypothesis of this paper is rather weak and not well described. For example, what is the link of LRO with critical brain hypothesis. For this testcase with VO2 array, what is the criticality and how is it derived? How is the criticality linked to phase dynamics of the VO2 array? These definitions are missing. What can one say when criticality occurs versus when LRO occurs? Coming to conclusion that criticality is not a prerequisite is a huge logical jump in the text without clear explanations.

13- Our classical machines do not operate in critical states but they do excel in some computational tasks. Thus, the point made in the conclusions is obvious. The interest of this paper +should be to show the different scale dynamics arising from thermal coupling and input voltage. What we learn from this and what can the reader exploit for neuromorphic systems, is rather left unanswered.

Point-by-point reply to the reviewers' comments

Reviewer #1 (Remarks to the Author):

In this manuscript, the authors have studied a computing platform based on a 2D grid of thermally coupled VO₂ oscillators. The authors have done a fine job analyzing the system's complex collective dynamics, highlighting phase transitions from an absence of oscillations to a state of synchronized oscillations by tuning the oscillator bias voltage and thermal capacitance. In particular, the authors reveal – via simulation and backed up by analytical analysis – a regime near the phase transition where the network has spatial long-range correlations. As shown by the authors, it is in this regime that the system has richer spatio-temporal dynamics and fading memory thanks to the existence of several time scales. Then, the authors simulate a 28x28 neuristor network as a reservoir for digit classification (MNIST) and observe that the system performs best in the region where oscillators are synchronized (rigid state), rather than near the phase transition region. From this experiment, the authors conclude that neither criticality nor long-range correlation is required for this particular task.

The well-written manuscript addresses an interesting topic of long-range correlation within complex dynamical systems which is very relevant for the neuromorphic and physics-based computing community.

However, my main concern is about the chosen experiment which I think should be further developed. In particular, I have the following comments:

1. Thanks to their internal fading memory, reservoir computers are generally used to process time series with the advantage that they do not require complicated training (compared to Recurrent NN). Why have the authors chosen to classify static data with a neuristor reservoir, rather than dynamical data? In my opinion, it is not so surprising that the reservoir performs best in the rigid state since the data is static, i.e. no memory is a priori needed to classify the data.

We thank the referee for this insightful observation. We selected the MNIST dataset for its ubiquity in benchmarking neural network performance, enabling straightforward comparative analysis across various computing approaches. Given the architectural design of our thermal neuristor array—emphasizing 2D spatial geometry and nearest-neighbor interactions—it is particularly suited for image processing tasks where spatial

correlations play a pivotal role. Acknowledging the referee's point on the potential advantages of dynamic data, we have now incorporated an additional experiment on the prediction of chaotic dynamics modeled by the 2D Kuramoto-Sivashinsky equation [Kalogirou15]. This experiment further elucidates the capabilities and limitations of our system under different operational conditions.

[Kalogirou15] A. Kalogirou et. al. (2015). An in-depth numerical study of the two-dimensional Kuramoto–Sivashinsky equation. *Proceedings of the Royal Society A: Mathematical, Physical and Engineering Sciences*, 471(2179), 20140932.

2. The authors could greatly improve the manuscript by performing a second experiment with dynamical data (such as spoken vowel recognition, time series prediction, etc.) and see if long-range order is helpful or not for this type of data.

We appreciate the referee's valuable suggestion to evaluate our model with dynamic data. Accordingly, we have introduced additional experiments using the 2D Kuramoto-Sivashinsky equation to simulate chaotic dynamics, leveraging the 2D spatial structure of our reservoir.

These experiments reveal that optimal performance is achieved using convolution-like, local 5-by-5-to-1 connections, underscoring the sufficiency of local connectivity for short-time predictions. Long-time evolution relies on global information, but in this setting, it is achieved by iteratively performing short-time predictions. We also attempted predicting long-time dynamics by directly utilizing long-range order from the reservoir, but this did not improve performance. In other words, long-range correlations are necessary for time series prediction tasks, but such correlations are not necessarily embedded within the reservoir.

3. I understand that using a Softmax output layer is convenient for having a probability distribution of classified labels. However, this is a highly non-linear stage that is cascaded after the reservoir. Hence, we cannot know for sure if the reservoir significantly contributes to the measured 96% accuracy (a 28x28 single-layer perceptron already achieves around 93%: <https://www.geeksforgeeks.org/single-layer-perceptron-in-tensorflow/>). Instead, I recommend the authors train the fully connected output layer with a linear regression – as it is usually done in reservoir computing, and obtain the output label with the Argmax function if needed.

We thank the referee for this insightful suggestion regarding the use of a softmax output layer in our model. Indeed, while softmax facilitates the interpretation of output

classifications as probability distributions, its non-linear nature could potentially mask the direct contributions of the reservoir to the observed classification accuracy.

In our system, the softmax function is applied after the fully connected layer, not after the reservoir output. Since softmax preserves the monotonicity of the output, substituting it with argmax post-training does not alter the classification outcomes, affirming the reservoir's contribution to achieving 96% accuracy. However, the integration of softmax with a negative log-likelihood loss significantly stabilizes training and enhances convergence, thereby optimizing performance.

To critically assess the impact of this non-linearity, we conducted additional experiments after removing the softmax transformation:

1. Linear Regression: Utilizing linear regression yielded 97% accuracy on the training set but only 88% on the test set, indicating a potential overfitting issue.
2. Adam Optimizer with MSE Loss: Employing mean squared error (MSE) loss and one-hot encoded labels resulted in 94% training accuracy and 93% test accuracy.

These experiments highlight the role of softmax in stabilizing and enhancing training outcomes, which is why we retained it in our primary experiments. We have updated the manuscript to elaborate on the rationale and functionality of including softmax in our model.

Moreover, in our experiments with the 2D Kuramoto-Sivashinsky equation where overfitting is less of a concern due to the ability to generate ample training data, we applied linear regression without additional non-linear transformations. This approach also demonstrated strong performance, reinforcing the versatility and robustness of our reservoir computing framework.

4. It would be interesting to compare the neuristor network with other existing work on thermally coupled VO₂ oscillators such as:

- A A Velichko et al 2020 IOP Conf. Ser.: Mater. Sci. Eng. 862 052062 (<https://iopscience.iop.org/article/10.1088/1757-899X/862/5/052062>)

- Velichko, A.; Belyaev, M.; Boriskov, P. A Model of an Oscillatory Neural Network with Multilevel Neurons for Pattern Recognition and Computing. *Electronics* 2019, 8, 75. <https://doi.org/10.3390/electronics8010075>

We thank the referee for directing our attention to these relevant studies involving thermally coupled VO₂ oscillators. These references are indeed relevant, as they explore mechanisms similar to those employed in our neuristor network, albeit with a focus on high-order synchronization for computational tasks.

Our approach diverges from the methodologies outlined in these papers mentioned above. While these studies leverage high-order synchronization of VO₂ oscillators to encode and process information, our system utilizes the inherent spiking dynamics of individual oscillators, supported by their thermal interactions, to facilitate computation. Specifically, our model capitalizes on both the local and long-range correlations induced by the thermal coupling to enhance the computational robustness and efficiency of the neuristor network.

To stress these points, we included a discussion of how our work relates to these studies, particularly focusing on the similarities and differences in the implementation and performance of thermally coupled VO₂ oscillators. This comparison will provide valuable context and demonstrate where our contributions stand within the field.

5. Could the authors discuss the pros and cons of using thermal exchanges compared to electrical connections? Is the neuristor network limited to nearest-neighbor connections?

Thank you for your inquiry regarding the use of thermal exchanges versus electrical connections in our neuristor network. The use of thermal interactions in our system primarily offers benefits in terms of energy efficiency and reduced fabrication complexity. Thermal interactions, by their nature, avoid the resistive losses common in electrical connections and simplify the manufacturing process by reducing the number of necessary circuit elements.

However, controlling thermal exchanges poses greater challenges compared to electrical signals, which can be precisely modulated. The diffusion of heat is slower and less localized, making it difficult to achieve tight control over the network's dynamics. Additionally, the network is currently limited to nearest-neighbor connections due to the nature of thermal diffusion. We are investigating ways to engineer the thermal conductance to enable long-range interactions, potentially increasing the network's connectivity and versatility.

6. What would be the impact of the external temperature on a neuristor network? Is the system robust to ambient temperature variations?

Thank you for this question regarding the impact of external temperature on our neuristor network. The operational stability of the network under varying ambient temperatures is indeed a critical factor for its practical application.

In our design, the external temperature is maintained at 325K (52°C), which ensures the optimal operation of the neuristor network. This setup mimics biological systems like the human brain, which also operate within a narrow temperature range for optimal functionality [Trastoy18].

We have observed that the network can tolerate slight deviations in the ambient temperature (a few Kelvins); however, these changes do influence the network's response characteristics. For instance, variations in temperature can affect the threshold voltages at which neuristors switch between insulating and conducting states due to the thermal sensitivity of the VO₂ material. This sensitivity means that while the network remains functional, its performance can shift with ambient temperature changes.

[Trastoy18] J. Trastoy, and I. K. Schuller. (2018). Criticality in the brain: evidence and implications for neuromorphic computing. ACS Chemical Neuroscience, 9(6), 1254-1258.

Reviewer #3 (Remarks to the Author):

Review of manuscript 'Collective Dynamics and long-range order in thermal neuristor networks'

Author present a novel spiking oscillator termed 'thermal neuristor' based on VO₂ devices. These devices are thermally coupled by sending heat spikes that perform image recognition tasks through reservoir computing.

The manuscript in the current provides several interesting concepts but in-depth descriptions are missing and overall it lacks clarity. Here are listed the comments.

1- VO₂ devices have been fabricated and used as the spiking oscillator that are thermally coupled. It is not clear how is thermal coupling established. What is the minimum distance between devices for thermal coupling to occur? Also, how are devices thermally coupled on lateral and vertical directions to nearest devices assuming that an array of devices is fabricated. What is the architecture and coupling scheme (fully or sparsely coupled) among devices for the reservoir computing? Even if performed in simulation such details are relevant.

We thank the referee for asking about the specifics of thermal coupling in our neuristor network. The thermal coupling mechanism is a critical aspect of our system, and we appreciate the opportunity to clarify these details further.

1. Establishment of Thermal Coupling: In our foundational work [Qiu23], we demonstrated heat propagation by observing resistance changes (ΔR) in adjacent VO₂ nanodevices when a neighboring device was powered. This powered device acted as a localized heat source, and the subsequent temperature changes in surrounding devices were detected via changes in their resistance, providing clear evidence of thermal coupling through the Al₂O₃ substrate. Capacitive or inductive interactions were also ruled out.
2. Minimum Distance for Effective Thermal Coupling: The minimum spacing required between devices to facilitate effective thermal coupling is influenced by factors such as the size of the devices and their power output. In [Qiu23], Figure 1b, we quantified the normalized resistance changes ($\Delta R/R_0$) in devices positioned at various distances from a heat-generating device, providing insights into the spatial extent of heat propagation. Additionally, the impact of device size on thermal coupling was investigated, revealing similar phenomena across different sizes [Qiu23].
3. Thermal Coupling Directionality: The heat generated within a nanodevice dissipates isotropically, affecting neighboring devices in both lateral and vertical directions in plane equally. This isotropic thermal propagation ensures uniform coupling across the array.
4. Device Array Configuration for Reservoir Computing: In our study, we utilized a 2D square lattice configuration for the array of VO₂ devices, with each device only interacting with its nearest neighbors. This setup is optimal for scaling up to larger arrays and can be practically implemented in a crossbar array architecture, as might be used in experimental setups.

[Qiu23] E. Qiu, et al. (2023). Stochastic transition in synchronized spiking nanooscillators. Proceedings of the National Academy of Sciences, 120(38), e2303765120.

2- Thermally coupled VO₂ devices have been investigated in the past. Thermal resistance coefficient to the substrate can have negative impact on heat buildup as heat transfer through the substrate (first shortest heat path) is faster than to the nearest device (secondary heat path). Thus, even if there is some heat coupling, it will be a weaker effect that might not cause a change in IMT threshold voltage. It could happen that some devices are stuck in insulating state unless enough voltage is pumped to them. Then, this also kills the purpose of having a thermally coupled network because via the voltage bias the switching dynamics are predominantly controlled for each spiking VO₂ oscillator. Thus, is there any understanding which is more predominant voltage or thermal coupling to the different phases that can be obtained. It could be that thermal coupling has a minor role and input voltage is the main key contributor to the phase dynamics.

We thank the referee for their comment regarding the interplay of thermal and voltage coupling in the collective dynamics of VO₂-based neuristor networks.

As documented in our recent publications [Qiu23] and [Qiu24], we have extensively explored the nature and impact of thermal interactions among neuristors. We intentionally choose a sapphire substrate for its high thermal and low electrical conductivity. This design ensures that the heat generated by the VO₂ spiking oscillators efficiently propagates through the sapphire substrate to the neighboring devices. In [Qiu23], we demonstrated robust thermal coupling due to the short separation distance (500 nm) between neighboring nanodevices. This strong thermal coupling is crucial, as shown in [Qiu24], where the VO₂ devices are biased with a subthreshold voltage, and it is the heat propagation that triggers the insulator-metal transition. Additionally, we observed a variety of neuronal functionalities, such as leaky-integrate-and-fire, synchronization, excitatory and inhibitory behaviors, all arising directly from the thermal coupling [Qiu24].

More importantly, this manuscript clearly illustrates that the emergent collective behaviors—such as the distinct phases of activity observed across the network—rely fundamentally on the thermal coupling. It is the thermal coupling that gives rise to the memory effects and long-range order in our system, driven by the distinct time scales as detailed in the analytical understanding section of our manuscript. Thus, we conclude that thermal coupling is not only essential but also dominant in influencing the collective dynamics of our neuristors, hence our designation of these devices as “thermal neuristors.”

To ensure a comprehensive understanding of these interactions, we have enhanced our manuscript with additional discussions [Qiu23] and [Qiu24], providing a more nuanced exploration of how thermal and voltage couplings collectively contribute to the phase dynamics of our neuristor network.

[Qiu23] E. Qiu, et al. (2023). Stochastic transition in synchronized spiking nanooscillators. *Proceedings of the National Academy of Sciences*, 120(38), e2303765120.

[Qiu24] E. Qiu et. al. (2024). Reconfigurable cascaded thermal neuristors for neuromorphic computing. *Advanced Materials*, 36(6), 2306818.

3- It is stated that VO₂ devices interact through heat. Can this be quantified and shown in a figure. For example, two devices coupled through heat and impact on spiking. The Y-axis can show the spike amplitude and x-axis can show the amount of heat. It will be also relevant to show the distance between two devices to achieve such amount of heat. Also,

the impact of thermal conductance to the substrate should be included. For example, in Eqn.2 it is not included the substrate thermal conductance.

We thank the referee for this suggestion to visually represent the interaction between thermally coupled VO₂ devices in our network. The dynamic relationship between heat input and spiking behavior has indeed been a focal point of our research, as detailed in our publications [Qiu23] and [Qiu24]. In these studies, we systematically explored how varying the heat input, controlled by the voltage applied to one neuristor, influences the spiking patterns of a nearby neuristor. Quantifying the effect of distance on the thermal influence between devices was also presented in [Qiu23].

In our theoretical model, each VO₂ device and its substrate are treated as a single unit with a uniform effective temperature. The thermal conductance—both inter-device and between the device and its environment—are considered constants, which simplifies the model for computational tractability. This approximation, while reducing complexity, has proven effective in capturing the essential dynamics of our neuristor network, as evidenced by the alignment of our model predictions with experimental observations reported in [Qiu24].

We have expanded the discussion in our manuscript to address these considerations.

[Qiu23] E. Qiu, et al. (2023). Stochastic transition in synchronized spiking nanooscillators. *Proceedings of the National Academy of Sciences*, 120(38), e2303765120.

[Qiu24] E. Qiu et. al. (2024). Reconfigurable cascaded thermal neuristors for neuromorphic computing. *Advanced Materials*, 36(6), 2306818.

4- For the square LxL lattice comprised of identical VO₂ thermal neuristors, it is stated that different patterns are observed with different input voltages. It is important to state (and not written in the paper), that while each oscillator is identical, each oscillator does not experience the same amount of heat. For example, the peripheral oscillators have will experience less heat than the internal oscillators in the lattice. It is because of these nonuniform thermal distributions that long-range corrections and different time scales are observed while bearing in mind that thermal transfer is a slower phenomenon than the impact of input voltage on oscillators.

We thank the referee for bringing attention to the issue of nonuniform thermal distribution within our LxL lattice of VO₂ thermal neuristors. The referee correctly pointed out that peripheral oscillators in an open lattice configuration could experience different thermal conditions compared to their internal counterparts, potentially impacting the observed dynamics and long-range correlations.

To investigate thoroughly this phenomenon, we conducted additional simulations with both open and periodic boundary conditions. While open boundaries naturally introduce edge effects leading to nonuniform heat distribution, periodic boundary conditions simulate an infinite tessellation, which mitigates these edge effects and provides a more uniform thermal distribution among the oscillators.

Our findings from these simulations reveal that the emergent patterns of long-range correlations and dynamic behaviors are consistent across both types of boundary conditions. This consistency suggests that the unique dynamical phenomena we observed are robust and not solely dependent on the nonuniformity introduced by boundary effects.

We have expanded the discussion in the supplemental material of our manuscript to include these considerations. This section now elaborates on how different boundary conditions influence thermal distributions and the resultant network dynamics. By addressing these aspects, we provide a deeper understanding of the factors influencing our system and reinforce the validity of our theoretical models in capturing the essential behavior of thermally coupled neuristor networks.

5- Different time scales are also reported in the paper which is relevant, though any simulations to portray this is missing. In addition adjusting thermal scales and thermal capacitance, will directly change the VO₂ spiking patterns and this is well reported in literature. The interest here is to see the collective behavior of heat and input voltage, which leads to different scale dynamics.

We thank the referee for this query regarding the interaction between thermal dynamics and input voltage, and their collective impact on the dynamics across different time scales in our VO₂ thermal neuristor network.

The interplay between heat and input voltage is indeed central to our study. We have meticulously explored how these elements influence the spiking patterns and phase dynamics within the network. This relationship is illustrated through the phase diagrams in Fig. 3(c) and (d), where we depict the avalanche size distributions under varying conditions. These diagrams help delineate the different operational phases of the network, highlighting the influence of both thermal and electrical parameters.

The thermal time scale, $\tau_{th} = R_{th}C_{th}$, is directly influenced by the thermal capacitance, indicating how changes in C_{th} affect the network dynamics. Similarly, the charging time of the capacitors, influenced by the input voltage, plays a crucial role in shaping the neuristor behavior.

To facilitate a deeper understanding of these dynamics, we have included additional simulation data in the supplemental materials. This data focuses on visualizing the collective spiking patterns across the phase diagram and illustrates how variations in electrical and thermal scales impact these patterns.

We appreciate the referee's feedback, which has prompted us to elaborate on these aspects comprehensively in our revised manuscript. We believe these additions will provide a clearer and more thorough understanding of the crucial dynamics at play in our neuristor network.

6- The analytical understanding section is weak (Section II.C). The different observed timescales are due to electrothermal effects and electrothermal simulations should reveal this (again missing any figure to portray this). It is not clear what authors define as 'memory' and 'long-range correlations'. These so-called 'long-range correlations' are the slow heat propagation impact on devices? An electrothermal simulation should also reveal this.

We appreciate this constructive feedback regarding Section II.C of our manuscript, particularly concerning the clarity needed in our discussion of "memory" and "long-range correlations" in our VO₂ neuristor network. We recognize the importance of clarifying these concepts more thoroughly and providing a clearer connection between our theoretical explanations and simulation results.

Regarding the referee's reference to "electrothermal simulations," our approach indeed integrates both electrical and thermal dynamics within a unified modeling framework, as detailed through Equations (1) and (2) in the main text. These equations capture the essential electrothermal interactions within the network, which are pivotal for our analyses. Although a detailed finite-element simulation could provide a more granular view of these interactions, such a level of detail exceeds the scope and intent of our current study, which aims to reveal the fundamental phenomena governing the system's behavior through more generalized models. This will be incorporated in our next study.

In response to the referee's comments, we have expanded Section II.C to better articulate what we define as "memory" within the context of our system. Here, "memory" refers to the ability of the network to retain long-term information about past states, resulting from its thermal inertia and electrical capacitance. This property is critical for understanding the network's temporal dynamics and its response to varying inputs.

Additionally, we clarify "long-range correlations" as the phenomena where the state of one neuristor influences others over macroscopic distances comparable to the size of the system. This effect is pivotal for the emergent behaviors observed in our network, including synchronization and phase transitions.

A more rigorous analytical understanding is provided in the supplemental material. Here, we provide a rigorous derivation showing how thermal and electrical dynamics interact over time to affect network behavior. Specifically, we demonstrate mathematically how the temperature of one neuristor can influence others located far away after a time proportional to the thermal diffusion timescale, τ_T . This derivation solidifies our explanation of long-range effects as a direct consequence of the electrothermal interplay within the neuristor network.

These revisions aim to provide a more robust theoretical foundation for our findings, linking our simulations closely with the theoretical analysis to offer a comprehensive understanding of the complex dynamics governing our neuristor network. We believe these additions will significantly strengthen the section and address the referee's concerns effectively.

7- Section II.D is also not clear and the link with the earthquake model dynamics is not sufficiently elaborated to understand the analogy. What does 'avalanche' represent in the context of VO2 array network? It is written that avalanche is a series of spiking events occurring at close spatial and temporal proximity. How are these defined such as close proximity, how close? Or temporal proximity, what's the timescale? If the aim is to verify the presence of long-range order effects due to heat, then the comparison of dynamics without heat vs with heat should reveal this (again electrothermal simulations should be useful here to provide insights). If the aim is to show the existence of long-range effects, a careful comparison should be used to see if indeed LROs are indeed due to the slow thermal phenomenon.

We thank the referee for valuable feedback on Section II.D and the need for greater clarity in our analogy to earthquake dynamics and the definition of avalanches within the VO2 array network. We recognize the importance of these explanations for understanding the complex behaviors observed in our study.

Definition of Avalanches:

In our model, an "avalanche" refers to a contiguous series of spiking events that occur in close spatial and temporal proximity within the network. Spatially, we define proximity as events involving nearest neighbors. Temporally, proximity is defined by a time window of 400 ns. This time window was chosen to be sufficiently small relative to the typical inter-spike interval, allowing us to effectively capture the discrete events that constitute a single avalanche. We have elaborated on this categorization process in the methods section, where we describe the construction of a 3D lattice (2D spatial and 1D temporal dimensions) to systematically analyze these events.

Analogy to Earthquake Dynamics:

The analogy to the Olami-Feder-Christensen earthquake model [Olami92] is based on the observed power-law distributions of avalanche sizes in our neuristor network, which are similar to those seen in earthquake dynamics. This model is particularly relevant because it also demonstrates long-range correlations without true scale invariance—characteristics that our network exhibits. We have refined our discussion of this analogy in the manuscript, providing a clearer explanation of how these power-law distributions relate to the underlying dynamics within our VO₂ array network.

Verification of Long-Range Order (LRO):

To explicitly verify the presence and origins of LRO in our network, we conducted additional simulations with and without thermal coupling. These comparative analyses revealed that in the absence of thermal interactions, the spikes across the network become uncorrelated, thus highlighting the essential role thermal dynamics plays to facilitate LRO. Results from these comparative simulations have been added to the revised supplemental materials, offering comprehensive evidence of the mechanisms through which LRO emerges in our network.

These enhancements and additions to the manuscript significantly improve the clarity and depth of our explanations, providing a more robust theoretical and empirical foundation for our findings.

[Olami92] Z. Olami, H. J. S. Feder, and K. Christensen. (1992). Self-organized criticality in a continuous, nonconservative cellular automaton modeling earthquakes. *Physical review letters*, 68(8), 1244.

8- What does figure 3 and its subfigures represent? The explanations on Section II.D are not clear and why power-law is not fitting and what should one understand if power law is fitting or not. Thus, it is not clear how power law on these graphs are used to reveal anything meaningful.

We thank the referee for pointing out the need for clearer explanations regarding Figure 3 and the significance of power-law distributions in our study. We recognize the importance of making these aspects of our analysis more accessible and have taken steps to clarify them in the manuscript.

Overview of Figure 3:

- Figure 3(a) and 3(b) display avalanche size distributions from our numerical experiments, where we observe a power-law behavior within certain parameter ranges. These figures are meant to illustrate how different operational conditions of the network can lead to variations in avalanche dynamics.

- Figure 3(c) presents a phase diagram that maps out the system's behavior across a range of input voltages and thermal capacitances. This diagram helps identify different operational phases based on the observed dynamics.
- Figure 3(d) depicts the exponents from the power-law fits to the avalanche size distributions. The fit of a power-law to these distributions across various system conditions indicates the presence of long-range correlations.

Significance of Power-Law Fits:

In systems exhibiting self-organized criticality, such as the sandpile model [Bak87] or neuronal networks [Beggs03], a power-law distribution of avalanche sizes is a signature of criticality, indicating that the system naturally evolves to a critical state characterized by scale invariance. In the context of our study, a power-law fit suggests the presence of long-range order within the neuristor network, implying that local perturbations can have widespread effects across the network.

To further assist readers in understanding these concepts, especially those who may not have a background in complex systems or statistical physics, we have included additional examples in the supplemental materials. These examples contrast cases where avalanche size distributions can and cannot be fitted with a power-law, illustrating the significance of such fits for the underlying dynamics of the system.

We hope that these additions and clarifications enhance the insights derived from Fig. 3 and demonstrate the significance of these findings in understanding the complex behavior of VO₂ neuristor networks.

[Bak87] P. Bak, C. Tang, and K. Wiesenfeld.(1987). Self-organized criticality: An explanation of the 1/f noise. *Physical review letters*, 59(4), 381.

[Beggs03] J. M. Beggs, and D. Plenz. (2003). Neuronal avalanches in neocortical circuits. *Journal of neuroscience*, 23(35), 11167-11177.

9- Authors write 'First, the power-law distributions in Fig. 3(a)(b) do not align with the finite-size scaling ansatz [32,33], which predicts diminishing finite-size effects with increasing system size.' This sentence is not related with the rest of what's written earlier in the paper, it reads disconnected from the rest and also it is not clear why is this relevant and what's the link with the topic at hand on different scale dynamics of VO₂ array network.

We thank the referee for pointing out the need for greater clarity regarding the discussion of finite-size scaling in our manuscript. We appreciate the opportunity to clarify the relevance of this concept to our findings and ensure a more cohesive narrative throughout the paper.

In the section we aimed to clarify the nature of long-range order (LRO) observed in our VO₂ neuristor network as distinct from the phenomena typically associated with criticality,

which are explored through finite-size scaling. Criticality involves a system at a critical point characterized by exact scale invariance, where power-law distributions are invariant under the rescaling of system size. Finite-size scaling is a technique used to confirm this by showing that finite-size effects diminish as system size increases, aligning data from different sizes to a universal scaling function.

However, our findings indicate that while our system displays power-law distributions suggestive of LRO, they do not conform to traditional finite-size scaling expectations. This suggests that our system maintains long-range correlations not due to traditional criticality but due to memory effects within the network dynamics, a phenomenon we term "memory-induced long-range order." This concept is crucial for understanding the unique dynamics within our system, where power-law behaviors are driven by embedded memory effects rather than a critical point.

To better integrate this discussion and clarify its relevance, we have revised the section on finite-size scaling in the manuscript. We explain explicitly how our observations of power-law fits and their deviation from finite-size scaling are instrumental in identifying the unique type of LRO driven by memory effects in our network. This helps differentiate our findings from classical critical phenomena and underscores the novelty of our study.

Furthermore, our group have been working on manuscripts that explore similar concepts in different systems [Sipling24], illustrating the broader applicability and significance of memory-induced long-range order. We believe these revisions will make the manuscript more coherent and provide a clearer understanding of how our findings contribute to the broader discourse on complex dynamical systems.

[Sipling24] C. Sipling, M. Di Ventra. (2024). Memory-induced long-range order in dynamical systems. arXiv preprint arXiv:2405.06834.

10- In the same sections, it is written 'Contrary to this, the smaller system sizes (32x32, 64x64) in Fig 3(a) results in the largest avalanches, contradicting finite-size scaling expectations'. Again, this is not clear, what is the expected finite size scaling? What's this about and what does it represent. These are unclear because section II.C and II.D are not clearly written or explained.

We thank the referee for their observations regarding the explanations of finite-size scaling and the reported behaviors of different system sizes in our manuscript. We recognize that the clarity of Sections II.C and II.D is crucial for understanding the significance of our findings, and we appreciate the opportunity to improve these sections.

Clarification on Finite-Size Scaling:

Finite-size scaling is a statistical physics approach used to understand how properties of small, finite systems approximate those of infinitely large systems as the size increases. This method is particularly valuable in studying systems near criticality, where it predicts that properties such as correlation lengths and fluctuation magnitudes (e.g., avalanche sizes in our context) should show universal behavior as system size increases. Typically, at criticality, larger systems would support larger avalanches due to longer correlation lengths.

Observations and Revisions:

Contrary to these typical expectations, our initial observations indicated that smaller systems (32x32 and 64x64) resulted in larger avalanches. This was puzzling as it suggested a deviation from expected finite-size scaling behavior at criticality, where larger systems should exhibit larger avalanches.

To address this, we revisited our analysis methods in the manuscript revision. We adjusted how avalanches are computed, ensuring each site can only contribute to an avalanche once, thereby preventing indefinitely large avalanches. We also switched to logarithmic binning for avalanche size distributions, which provided a more accurate representation.

Updated Findings and Implications:

Even with these methodological adjustments, our subsequent analyses (detailed in Fig. 9 of the supplemental materials) still show that rescaling of avalanche size distributions does not collapse the data into a unified finite-size scaling ansatz. This outcome further confirms that our system does not exhibit traditional critical behavior under finite-size scaling. Instead, it supports our hypothesis that the system demonstrates memory-induced long-range order, driven by thermal memory effects within the VO₂ network, rather than classical criticality.

We have revised Sections II.C and II.D to provide a clearer and more comprehensive explanation of finite-size scaling and its relevance to our findings. We clarify why the deviations from expected scaling behavior are significant and discuss how these observations support the existence of memory-induced long-range order in our system. This type of long-range order does not scale with system size as expected in critical systems because it is influenced by different underlying dynamics—namely, the thermal interactions and memory effects in the neuristor network.

11- Section III, experiments were performed varying thermal capacitance C_{th} and noise strength. How realistic would be to varying C_{th} knowing that this is a material property. For example, depending on the fabricated VO₂ devices, the thermal resistance and capacitance can be obtained. In addition, what is noise in this network, and how is it introduced?

We thank the referee for their inquiry regarding the experimental manipulation of thermal capacitance and the nature of noise in our VO₂ neuristor network.

In an experiment, it is possible to modulate C_{th} by varying the physical characteristics of the device, such as changing the thickness or material composition of the substrate. For example, using a thicker substrate or a material with higher specific heat can increase the thermal capacitance. This approach allows us to explore the dynamics of the neuristor network under different thermal conditions, although the range of feasible variations might be limited by practical material and fabrication constraints.

Noise in our neuristor network is introduced by both intrinsic and extrinsic factors, which contribute to the system's complexity and realistic behavior:

- The random distribution of metallic and insulating domains within the VO₂ contributes to variability in thermal conductance, which affects the consistency of thermal propagation and introduces jitter in the spiking behavior of the neuristors.
- Fluctuations in heat generation during current spikes lead to variability in heat propagation, enhancing the stochastic nature of the network's dynamics.
- Imperfections in the VO₂ thin films and damage to the sapphire substrates caused by etching processes create non-uniformities in phonon propagation. These defects can significantly impact the thermal behavior of the system, adding to the overall noise level.

Secondly, what is meant by “identifying the region of phase spaces would produce optimal result”, please define these.

The term "phase space" in our context refers to a diagrammatic representation where each point corresponds to a particular set of system parameters (like C_{th} , input voltage, and noise strength). These diagrams (as shown in Figs. 3 and 6) are crucial for visualizing how different parameter combinations affect the system's behavior. Identifying regions within these phase spaces that yield optimal results involves determining the parameter sets that maximize performance metrics, such as classification accuracy. Our goal is to fine-tune these parameters to enhance the overall effectiveness and reliability of the network in practical applications.

We have expanded and clarified these explanations in the revised manuscript to ensure that the methodology and its implications are accessible and comprehensible to all readers, regardless of their familiarity with the technical aspects of neuromorphic systems.

It is expected that the system would show different behavior and phase dynamics when voltage is varied. Thus, it is not clear what the authors want to show via the RC with VO₂ arrays with respect to long-range order correlations (as stated in earlier part of the paper). It is expected that RC will provide rich phase dynamics, but what are authors aiming at to link between RC about presence/or not presence of LRO? This is not clear. Because, regardless of LRO, RC will perform some computation and extract information from highly nonlinear neurons, as this is also the aim of RC, thus, RC computation without LRO is not a surprise. For example, as a comparison proof, the same VO₂ network with electrical coupling but without thermal coupling can be used for RC.

Our research primarily explores the dynamics of RC implemented with VO₂ neuristor arrays, particularly focusing on the role of LRO in computational efficiency. Commonly, it is hypothesized, especially in theories like the critical brain hypothesis and concepts surrounding the "edge of chaos," that LRO and near-critical dynamics are crucial for maximizing the computational capabilities of neuromorphic systems. These theories suggest that such conditions enable a system to access a wider range of dynamical states, potentially enhancing its ability to process complex information.

In our study, however, we demonstrate that efficient computation in RC systems can occur without the presence of LRO. We have specifically demonstrated that optimal performance is achieved within a rigid state of the VO₂ neuristor arrays, thereby challenging the universal criticality requirement for optimal computational performance. Furthermore, we have incorporated an additional RC experiment in this revision, which involves predicting chaotic dynamics governed by the 2D Kuramoto-Sivashinsky equation. This experiment further supports our conclusion that LRO is not necessary for successful RC. By demonstrating that VO₂-based RC systems can function effectively without reliance on long-range correlations, we provide a basis for reevaluating the design and theoretical underpinnings of future neuromorphic architectures.

12- The main hypothesis of this paper is rather weak and not well described. For example, what is the link of LRO with critical brain hypothesis. For this testcase with VO₂ array, what is the criticality and how is it derived? How is the criticality linked to phase dynamics of the VO₂ array? These definitions are missing. What can one say when criticality occurs versus when LRO occurs? Coming to conclusion that criticality is not a prerequisite is a huge logical jump in the text without clear explanations.

We thank the referee for the critical feedback regarding the foundational findings of our paper. We recognize the importance of clearly defining and discussing the concepts of long-range order (LRO), criticality, and their relationship to the critical brain hypothesis. We appreciate the opportunity to enhance our explanations and strengthen the logical framework of our manuscript.

Our research investigates the dynamics of a VO₂ neuristor network to explore the manifestations of long-range order (LRO) and its relationship to criticality—a concept central to theories like the critical brain hypothesis. In this hypothesis, criticality is often linked with optimal computational performance, positing that being near a critical point enables a system to maximize information processing capabilities.

In our study, we demonstrate that the VO₂ neuristor array exhibits LRO, characterized by the occurrence of power-law distributions in the avalanche sizes. Importantly, these distributions lack scale invariance, a hallmark of true criticality. This finding is significant because it shows that while our system exhibits LRO, it does not conform to the traditional definition of a critical system. This challenges the assumption that LRO must inherently be linked to criticality.

Further, our numerical experiments provide compelling evidence that efficient computation within this neuristor network does not require the system to be at or near a critical state. More crucially, we show that such computation can even occur in the absence of LRO. By comparing computational performance under conditions with and without LRO, our results indicate that the network can achieve high computational efficiency without relying on long-range correlations. This finding prompts a reevaluation of the need for LRO or criticality in designing effective neuromorphic systems.

These insights significantly impact the theoretical framework for neuromorphic computing. They suggest that the pursuit of critical-like states or the engineering of systems to exhibit LRO might not be necessary for achieving high computational performance in some tasks. In response to your feedback, we have revised the manuscript to include a clearer exposition of these points, providing a detailed discussion on how our findings relate to the broader discourse on neuromorphic system design. We have also refined the logical flow to more explicitly link our empirical evidence with these theoretical implications, ensuring that the conclusions drawn are well-supported and comprehensible.

We hope that these revisions will address the concerns raised and more clearly communicate the significance of our findings, highlighting our contribution to the understanding of neuromorphic computing dynamics.

13- Our classical machines do not operate in critical states but they do excel in some computational tasks. Thus, the point made in the conclusions is obvious. The interest of this paper should be to show the different scale dynamics arising from thermal coupling

and input voltage. What we learn from this and what can the reader exploit for neuromorphic systems, is rather left unanswered.

We thank the referee for encouraging us to explain further the significant aspects of our study. This feedback allows us to highlight the depth of the dynamics observed in our VO₂ neuristor networks and how these insights can be applied to neuromorphic computing. It also prompts a more detailed discussion on the implications of challenging the well-received hypothesis regarding criticality and computational efficiency.

Our research reveals the rich dynamical behavior of VO₂ neuristor networks, primarily driven by the intricate interplay between thermal coupling and input voltage. These dynamics include multiple stable and metastable states, transitions between different phases of activity, and the emergence of complex spatiotemporal patterns. Each of these dynamical states can potentially be harnessed for different computational tasks, offering a palette of behaviors that can be selectively activated through careful tuning of the system parameters.

For neuromorphic computing, this means that a single hardware platform, through parameter modulation, can emulate various types of neural behaviors seen in biological systems. For example, the ability to shift between different dynamic states allows for adaptive computing, where the system can change its processing strategy based on the input characteristics or desired output. This adaptability could be particularly useful in real-time processing environments or tasks that require a high degree of sensitivity to input variability.

The hypothesis that systems operating at or near a critical point—characterized by long-range correlations and criticality—can achieve superior computational efficiency is prevalent. This hypothesis is rooted in the belief that critical systems can maximize information capacity and adaptivity, a concept extrapolated to neural dynamics in the critical brain hypothesis.

Our study challenges this prevailing view by demonstrating that VO₂ neuristor networks can achieve high levels of computational performance without exhibiting criticality or even significant long-range order. The implication here is profound: it suggests that the pursuit of criticality may not be necessary for designing effective neuromorphic systems. Instead, our findings advocate for a broader exploration of non-critical dynamical regimes that might offer computational capabilities just as powerful, if not more so, than those found at criticality.

Considering these discussions, we have revised our manuscript to articulate more clearly how the rich dynamics observed in our study can be leveraged for neuromorphic computing applications. We also elaborate on how our findings challenge the necessity

of criticality for computational efficiency, proposing that future research should explore a wider array of dynamical regimes. By doing so, we aim to broaden the design and operational paradigms for neuromorphic systems, potentially leading to more versatile and robust computing architectures.

We trust that these elaborations address the criticism of the referees comprehensively and clarify the transformative implications of our research for neuromorphic computing. We are excited about the potential of our findings to influence future studies and practical applications in the field.

References

- [Bak87] P. Bak, C. Tang, and K. Wiesenfeld.(1987). Self-organized criticality: An explanation of the $1/f$ noise. *Physical review letters*, 59(4), 381.
- [Beggs03] J. M. Beggs, and D. Plenz. (2003). Neuronal avalanches in neocortical circuits. *Journal of neuroscience*, 23(35), 11167-11177.
- [Kalogirou15] A. Kalogirou et. al. (2015). An in-depth numerical study of the two-dimensional Kuramoto–Sivashinsky equation. *Proceedings of the Royal Society A: Mathematical, Physical and Engineering Sciences*, 471(2179), 20140932.
- [Kuramoto78] Y. Kuramoto. (1978). Diffusion-induced chaos in reaction systems. *Progress of Theoretical Physics Supplement*, 64, 346-367.
- [Olami92] Z. Olami, H. J. S. Feder, and K. Christensen. (1992). Self-organized criticality in a continuous, nonconservative cellular automaton modeling earthquakes. *Physical review letters*, 68(8), 1244.
- [Qiu23] E. Qiu, et al. (2023). Stochastic transition in synchronized spiking nanooscillators. *Proceedings of the National Academy of Sciences*, 120(38), e2303765120.
- [Qiu24] E. Qiu et. al. (2024). Reconfigurable cascaded thermal neuristors for neuromorphic computing. *Advanced Materials*, 36(6), 2306818.
- [Sipling24] C. Sipling, M. Di Ventra. (2024). Memory-induced long-range order in dynamical systems. *arXiv preprint arXiv:2405.06834*.
- [Trastoy18] J. Trastoy, and I. K. Schuller. (2018). Criticality in the brain: evidence and implications for neuromorphic computing. *ACS Chemical Neuroscience*, 9(6), 1254-1258.

REVIEWER COMMENTS

Reviewer #1 (Remarks to the Author):

The authors have greatly improved the manuscript. It reads better and the research objectives are clearer. I also thank the authors for adding a time series prediction experiment which I find very elegant. I find the LRO topic very interesting and relevant for the neuromorphic community. However, I think the manuscript lacks clear evidence compared to the strong conclusions, and I recommend the authors add more quantitative results from their two main experiments to convince the reader. In particular, I have the following comments:

1) Authors conclude that for the MNIST classification task, the system performs better in the rigid state and LRO is not required. However, for the selected parameters, it is not obvious the system operates in the rigid phase. From Fig.10a in SM, it seems that the chosen parameters correspond to a point in or at the edge of the LRO region, and not in the rigid state. Since the authors state this is a key finding, they should clearly show evidence and highlight it in the main paper. For instance, Fig.3 could be adapted to the noise level selected for the MNIST experiment ($\sigma=0.2$), highlighting the operating region [$V_{min}=10.5V$; $V_{max}=12V$] set for the experiment. However, this would assume that all neuristors have the same input (either V_{min} or V_{max}), whereas in the experiment, inputs are a combination of V_{min} and V_{max} according to the input image. Therefore, in addition to the updated Fig.3, authors must show the corresponding avalanche distribution measured during the MNIST experiment, since it is the metric used to quantify LRO throughout the paper. This quantitative result (or any other kind proving the absence of LRO) is also missing from the second experiment predicting the 2D KS dynamics.

2) In Fig.2 and video 1 of SM, the authors associate the rigid state with oscillations in the system suggesting global synchronization. I understand this contrasts with long propagating waves (avalanches) occurring with LRO. However, in video 2 of the MNIST experiment, it seems that the reservoir has some “correlated clusters” after some time that look like small avalanches, like the $V_{in}=10V$ case of video 1. Authors should clarify this point with more quantification, as suggested by my comment 1).

3) I recommend adding a plot in Figure 9 of SM that shows a typical avalanche probability distribution for a scale-invariant example. In the inset Fig.9a), it is not clear at first sight whether the curves “overlap well” or not, and there is no quantification of such “overlap”. While this may be obvious for the authors, they should compare with a scale-invariant example to convince the reader who is not necessarily familiar with criticality and scale-invariance properties.

4) Authors explain the presence of LRO due to the interplay of slow/fast dynamics and conclude that LROs are not useful for certain tasks such as digit recognition or time series prediction, since LROs seem absent during the experiments. To add strength to their conclusions, I think it would be very interesting to check if having similar time scales removes LROs and if it produces similar results. I am aware that removing the slow time scale might impoverish the dynamics and the accuracy, regardless of whether having LROs or not. However, I believe it is worthwhile trying since this could constitute a double verification of the authors' claims and would have important practical implications when operating the hardware: inference could be then executed much faster. To that end, I suggest repeating the MNIST classification experiment by biasing the VO2 devices in the middle of its negative differential resistance region with $R_{load} \sim R_{met}$ (and adjusted V_{dd}) so that the capacitor charge/discharge times are both similar to the thermal time constant.

Sincerely,

Corentin Delacour

Reviewer #3 (Remarks to the Author):

The revised manuscript has addressed all my concerns and it can be considered for acceptance.

REVIEWER COMMENTS

Reviewer #1 (Remarks to the Author):

The authors have greatly improved the manuscript. It reads better and the research objectives are clearer. I also thank the authors for adding a time series prediction experiment which I find very elegant. I find the LRO topic very interesting and relevant for the neuromorphic community. However, I think the manuscript lacks clear evidence compared to the strong conclusions, and I recommend the authors add more quantitative results from their two main experiments to convince the reader.

Thank you for your constructive feedback and positive remarks regarding the improvements to our manuscript. Your suggestions on verifying and quantifying LRO in RC tasks have been instrumental in improving our manuscript, completing our arguments and strengthening our claims.

In particular, I have the following comments:

1) Authors conclude that for the MNIST classification task, the system performs better in the rigid state and LRO is not required. However, for the selected parameters, it is not obvious the system operates in the rigid phase. From Fig.10a in SM, it seems that the chosen parameters correspond to a point in or at the edge of the LRO region, and not in the rigid state. Since the authors state this is a key finding, they should clearly show evidence and highlight it in the main paper.

Thank you for your astute observation regarding the operational phase of the system in our MNIST classification task. The very first version of Fig. 10a (attached below) did accurately depict the chosen parameters within the rigid state. However, subsequent updates to Fig. 10a, prompted by the revised avalanche binning criteria, inadvertently obscured this crucial detail, leading to an unclear delineation of the LRO boundary. To address this, we have conducted additional simulations across the parameter range $V \in [10.5V, 12.2V]$ and $\sigma = 0.2\mu] \cdot s^{-\frac{1}{2}}$, which confirm that these conditions indeed correspond to the rigid state. We have revised Fig. 10a to more accurately reflect this and have added further snapshots of the spiking activities under these settings in Fig. 12b to provide clear visual evidence supporting our conclusions.

For instance, Fig.3 could be adapted to the noise level selected for the MNIST experiment ($\sigma=0.2$), highlighting the operating region [$V_{\min}=10.5\text{V}$; $V_{\max}=12\text{V}$] set for the experiment.

Thank you for your suggestion regarding the adaptation of Fig. 3 to better reflect the noise level and operating region used in our MNIST experiment. According to our observations in Fig. 10a, at a noise level of $\sigma = 0.2\mu\text{J} \cdot \text{s}^{-\frac{1}{2}}$ and $V = 12\text{V}$, the system predominantly operates in the rigid state or exhibits no activity, with only a minimal region displaying LRO. The occurrence of LRO becomes even less prevalent as the noise level decreases. In the initial sections of our paper, we aim to showcase the diverse dynamics within the system, which is why we opted to maintain a higher noise level of $\sigma = 1\mu\text{J} \cdot \text{s}^{-\frac{1}{2}}$. This setting ensures a broader range of observable activities within the system, thereby enriching our analysis.

However, this would assume that all neuristors have the same input (either V_{\min} or V_{\max}), whereas in the experiment, inputs are a combination of V_{\min} and V_{\max} according to the input image. Therefore, in addition to the updated Fig.3, authors must show the corresponding avalanche distribution measured during the MNIST experiment, since it is the metric used to quantify LRO throughout the paper. This quantitative result (or any other kind proving the absence of LRO) is also missing from the second experiment predicting the 2D KS dynamics.

Thank you for your constructive feedback. As you correctly noted, assuming a rigid state with uniform inputs does not directly translate to the same state under varied inputs, such as those derived from the patterns of handwritten digits. To address this, we conducted

further experiments to quantify avalanche distributions during both the MNIST prediction tasks and the 2D KS dynamics experiments.

Our results show that for the MNIST task, the avalanche size distribution reveals additional long-range structures distinct from the previously observed LRO, whereas for the KS dynamics, the avalanches are predominantly system-wide, indicative of a rigid state. Superficially, these observations may seem to indicate that the additional structure in the MNIST dataset's avalanche size distribution does not support our prior conclusions. However, these long-range structures originate from the dataset rather than from intrinsic dynamics of our neuristor network.

To show this explicitly, we implemented two additional experiments:

1. We removed the thermal coupling between the neuristors, completely eliminating correlations within the reservoir. After optimizing the hyperparameters, our reservoir computing (RC) algorithm achieved a 95.8% accuracy on the MNIST dataset. Without interactions in the reservoir, the corresponding avalanche size distribution mirrors the structure of the dataset itself.
2. Following your suggestion in question 4), we also conducted an experiment to reduce the presence of LRO by significantly decreasing the slower time scale. This adjustment led to an RC algorithm accuracy of 94.2%, with an avalanche size distribution similar to the original setup, still reflecting long-range structures inherent to the MNIST dataset, not in the neuristor network. Given the difficulty for this modified reservoir to maintain LRO, we infer that the observed long-range structures are derived from the dataset.

Across all three experiments, we observed no clear correlation between LRO and computational performance: while the original setup, which does not exclude LRO, performed best, the non-interacting setup achieved nearly comparable results without any long-range interactions. The experiment with reduced memory showed the lowest performance yet displayed similar long-range structures to the original. Consequently, as discussed in our work, we conclude that LRO is not necessary for effective computational performance in these contexts.

2) In Fig.2 and video 1 of SM, the authors associate the rigid state with oscillations in the system suggesting global synchronization. I understand this contrasts with long propagating waves (avalanches) occurring with LRO. However, in video 2 of the MNIST experiment, it seems that the reservoir has some "correlated clusters" after some time that look like small avalanches, like the $V_{in}=10V$ case of video 1. Authors should clarify this point with more quantification, as suggested by my comment 1).

Thank you for your meticulous observation regarding the dynamics observed in Supplementary Video 2 of the MNIST experiment. As discussed in response to your previous question, we have analyzed the avalanche size distribution with the MNIST dataset as input. This analysis confirms that the long-range structures, which appear as "correlated clusters," originate from the dataset itself.

Following the implementation of the two additional experiments, we have updated Supplementary Video 2, and included Videos 3 and 4, which visualize the reservoir dynamics under the three different settings. These enhancements aim to provide clearer visual evidence and further quantify the phenomena discussed, addressing the points you have raised.

3) I recommend adding a plot in Figure 9 of SM that shows a typical avalanche probability distribution for a scale-invariant example. In the inset Fig.9a), it is not clear at first sight whether the curves "overlap well" or not, and there is no quantification of such "overlap". While this may be obvious for the authors, they should compare with a scale-invariant example to convince the reader who is not necessarily familiar with criticality and scale-invariance properties.

Thank you for your valuable suggestion to clarify the scale-invariance properties in our study. We appreciate the opportunity to enhance the reader's understanding of these concepts. Below, we present an example of avalanche size distributions from a spin-glass-like system with varying system sizes, as reported in [Sipling24], another recent study from our group. In this example, the curves demonstrate good overlap, indicative of scale invariance in the system.

Comparing this to Fig. 9 in our manuscript, it becomes apparent that our system does not exhibit similar scale invariance. To address this point more explicitly in the manuscript, we have revised the description in the supplementary materials to better explain the criteria for determining scale invariance. While we considered including a figure from an unrelated system to illustrate ideal overlap, we ultimately decided against it to avoid potential confusion. Instead, we have added references to literature that provide clear examples of scale-free systems, which should assist readers unfamiliar with these concepts.

[Sipling24] Sipling, C., and Di Ventra, M., "Memory-induced long-range order in dynamical systems." arXiv preprint arXiv:2405.06834 (2024).

4) Authors explain the presence of LRO due to the interplay of slow/fast dynamics and conclude that LROs are not useful for certain tasks such as digit recognition or time series prediction, since LROs seem absent during the experiments. To add strength to their conclusions, I think it would be very interesting to check if having similar time scales removes LROs and if it produces similar results. I am aware that removing the slow time scale might impoverish the dynamics and the accuracy, regardless of whether having LROs or not. However, I believe it is worthwhile trying since this could constitute a double verification of the authors' claims and would have important practical implications when operating the hardware: inference could be then executed much faster. To that end, I suggest repeating the MNIST classification experiment by biasing the VO₂ devices in the middle of its negative differential resistance region with $R_{load} \sim R_{met}$ (and adjusted V_{dd}) so that the capacitor charge/discharge times are both similar to the thermal time constant.

Thank you for this insightful suggestion, which indeed has significant implications for the practical application of our hardware. Following your recommendation, we attempted to equalize the time scales by minimizing the insulating RC time. Specifically, we adjusted the ambient temperature settings to reduce the insulating resistance of the VO₂ devices. This approach ensured that the neuristors started at a lower resistance and returned to this state more rapidly after each spiking event.

Our results indicate that completely removing the slower time scale will cause the neuristors to stop spiking, rendering the system unable to perform information processing tasks. However, we could reduce the insulating RC time, τ_{ins} , from $7.57 \mu s$ to approximately $1 \mu s$, while still maintaining stable spiking dynamics. Although this modified time scale is approximately five times longer than the faster metallic RC time, it significantly diminishes the memory capacity within the system. Under these modified conditions, the spiking frequency increased, and the amplitudes of the spikes decreased.

Upon conducting the MNIST classification experiment with these settings, we achieved an accuracy of 94.2%. Notably, we observed long-range structures in the avalanche size distribution, which, as discussed in response to your first question, are inherent to the MNIST dataset rather than a result of LRO within the system. These findings support our

conclusion that there is no direct correlation between LRO and enhanced computational performance.

Reviewer #3 (Remarks to the Author):

The revised manuscript has addressed all my concerns and it can be considered for acceptance.

Thank you for your feedback and for acknowledging the revisions made to the manuscript. We appreciate your support and are pleased to hear that the manuscript now meets your expectations. We are grateful for the opportunity to improve our work based on your valuable insights.

REVIEWERS' COMMENTS

Reviewer #1 (Remarks to the Author):

The authors have addressed all my comments and I am happy with the revised manuscript.